# Emotional Processing and Psychological Well-Being of Healthcare Workers During the COVID-19 Pandemic

**DOI:** 10.3390/healthcare13080871

**Published:** 2025-04-11

**Authors:** Alessandra Busonera, Marco Tommasi, Ilenia Piras, Maura Galletta

**Affiliations:** 1Department of Medical Sciences and Public Health, University of Cagliari, 09042 Cagliari, Italy; alessandrabusonera@yahoo.it; 2Department of Medicine and Aging Sciences, G. d’Annunzio University of Chieti-Pescara, 66100 Chieti, Italy; marco.tommasi@unich.it; 3Emergency Department, SS. Trinità Hospital, ASL Cagliari, 09122 Cagliari, Italy; ilenia_78it@yahoo.it; 4Department of Medical Sciences and Public Health, Hygiene Section, University of Cagliari, 09042 Cagliari, Italy

**Keywords:** Italian healthcare workers, emotional processing, COVID-19 pandemic, work–life balance

## Abstract

**Objectives:** The present cross-sectional web-based survey study aimed to examine the impact of gender and frontline/non-frontline condition on psychological health (depression and peritraumatic distress) of a sample of Italian healthcare workers, and to explore the mediating effects of emotional processing in these relationships. **Methods:** Emotional processing styles and depressive and peritraumatic distress symptoms were assessed between March and December 2021, a period which in Italy corresponds to the third and the start of the fourth wave of COVID-19, along with gender, frontline versus non-frontline role in the pandemic emergency, and other sociodemographic and background variables. **Results:** Results showed that emotional processing mediated the relationship between gender and frontline/non-frontline role with depressive and peritraumatic distress symptomatology. Being a woman, working on the frontlines, and having a maladaptive emotional processing style increased the negative effects of stress during the pandemic. **Conclusions:** The obtained results strengthen the importance of providing useful psychological support for health professionals to facilitate the expression and management of emotions as well as the learning of healthy and functional styles of emotional processing.

## 1. Introduction

The coronavirus pandemic has represented an international health, economic, and social emergency that most Western countries were not prepared to deal with.

Italy was the first European country to experience the spread of SARS-CoV-2, beginning in January 2020 with a couple of tourists from Wuhan, China, followed by the first local cases in February [1]. As of February 2025, Italy has reported 27,192,437 cases, with 513,865 among healthcare workers (Italian National Institute of Health, https://www.iss.it/coronavirus, accessed on 21 February 2025). The spread was uneven, with northern regions more affected than the south, including Sicily and Sardinia. Sardinia, characterized by geographical isolation, limited connections, and low population density, seemed to have some protective factors. However, its aging population and strong tourism industry made it vulnerable, especially during summer, when tourist influxes exacerbated infection spread. Sardinia’s pandemic timeline mirrored Italy’s, with a lockdown starting in March 2020 and a significant second wave in October 2020. The third wave (February–May 2021) saw Sardinia’s situation align with the rest of Italy, aided by vaccination efforts. In the summer of 2021, tourist movement led to another peak, followed by a fifth wave (November 2021–February 2022) driven by the Omicron variant, which was less severe but highly transmissible [2].

Similarly to what happened in the rest of Italy, the lockdown began on the island in mid-March 2020, which resulted in the suspension of sea and air connections. On the contrary, the second pandemic wave (October 2020–January 2021) began earlier in Sardinia compared to other Italian regions, with an additional peak in mid-summer 2020 (linked to the movement of tourists) and a prolonged autumn-winter peak. The third wave of the infection (February–May 2021) saw the situation in Sardinia more similar to that of the rest of Italy, presumably due to the positive effect of the population’s participation in the vaccination campaign. During this period, characterized by the spread of the so-called Delta variant, the island recorded the lowest incidence of cases in Italy. However, in the immediately following period (June–October 2021), with the return of tourist flows, the island experienced another peak in cases, entering the fourth wave. The fifth and final significant wave (November 2021–February 2022) was characterized by the spread of the Omicron variant, which was less virulent than the previous one but highly transmissible, leading to a new significant increase in COVID-19 cases and quickly becoming dominant [2].

The coronavirus pandemic has mobilized a response from the health system involving thousands of HCWs around the world, who have seriously risked their health in taking an active part in managing the emergency. In addition to being among the workers at greatest risk of exposure to the virus, HCWs were exposed to operational and emotional overload [3,4].

Since 2020, numerous studies worldwide have shown that HCWs were at high risk for their mental health during the pandemic, highlighting a number of factors that can contribute to the increase in their psychophysical stress. Such risk factors include: exposure to the virus, poor preparation for emergency management, frequent exposure to patient death, fear of being infected and infecting loved ones, the experimentation of a sense of helplessness in the face of patients’ worsening conditions, the social stigma linked to a greater exposure to the disease, and changes in work related to organizational, relational, and safety aspects [5,6,7,8]. The forms of psychological distress most identified in HCWs during the pandemic included high levels of generalized, acute, and COVID-19-related anxiety [9,10,11,12], as well as high levels of depressive symptoms [11,13], stress [9,14,15], and posttraumatic symptomatology [16,17,18]. Other problems frequently found concerned the fear and worry related to COVID-19 (of becoming infected, infecting family members, dying, etc.), insomnia and other sleep disorders, burnout [12,13,19], and alcohol use disorders [20]. Regarding the HCWs’ psychological well-being protective factors, studies highlighted the role of higher perceived social and organizational support, personal resilience, and adaptive coping [15,20,21]. The most frequently reported risk factors for HCWs’ psychological distress include being a frontline worker, particularly if working in an emergency department; having concerns about infecting family members [19]; and a maladaptive coping style [10,11,15,22]. Furthermore, marked gender differences emerged, underlining how being a female HCW was a risk factor for greater psychological distress and worsened mental health during the pandemic [11,21,22,23,24].

However, there is a lack of studies focused on the role that HCWs’ emotional functioning plays in maintaining an adaptive level of psychological well-being in the pandemic context. This shortcoming is quite surprising when one considers the extensive literature that emphasizes the importance of emotional aspects for this population of professionals, especially in emergency situations [25,26].

In the context of the COVID-19 pandemic, the close and daily contact with suffering people, dealing with illness and death, and, in general, caring for patients inevitably involves the emotional sphere, eliciting various feelings in HCWs [27]. Adequate emotional competence is necessary for these professionals to avoid the risk of taking refuge in one of these two extremes: aseptic neutrality or being overwhelmed by patients’ feelings.

According to the emotional processing model (EPM), in order to successfully cope with stressful events, individuals must be able to recognize their own emotions and know how to express them in an adaptive way [28,29,30]. Coping failures may occur when, for example, people tend to suppress or avoid their emotions, as this inhibits emotional processing. Rachman [29] introduced the concept of *emotional processing* by defining it as “a process whereby emotional disturbances are absorbed and decline to the extent that other experiences and behavior can proceed without disruption” (p. 51). He points out that a majority of individuals successfully process the vast majority of disturbing events that occur in their lives, yet some individuals fail in processing emotions. If an emotionally disturbing event is not absorbed satisfactorily, some signs may become evident; they can recur intermittently and can be explicit and obvious or implicit and subtle [31]. In general, according to the EPM, a poor emotional processing capacity can prepare the ground for the development of psychological distress, medically unexplained somatic symptoms, and psychopathology [28,30,32,33]. Deficiencies in emotional processing, including increased rumination, avoidance, and maladaptive coping, have been found to be associated with psychiatric conditions such as Post Traumatic Stress Disorder (PTSD) [34,35], depression [36,37], anxiety disorders [38], and substance abuse disorders [35].

Emotional processing may help to explain the different impact of stress on different individuals. By applying this theoretical framework to the situation experienced by HCWs in the COVID era, we could hypothesize that an adequate capacity for emotional processing can be extremely important in safeguarding the psychological health of operators. PTSD and major depressive disorder (MDD) are two of the most common mental health conditions that can follow exposure to a highly stressful and/or traumatic event. These disorders share some fundamental disease processes and are frequently found to be comorbid [36,39]. Being able to promptly identify symptoms of psychological suffering, such as depression and peri-traumatic distress, can be very important to predict and prevent the risk of developing more serious psychopathology, such as PTSD, major depressive disorder (MDD), reactive depression, or other mood-related disorders in subsequent periods.

Based on the emotional processing model, on the theoretical relationships between the measured variables, and on the results already published in the literature, the main purpose of the current study was to examine, in a sample of Italian HCWs interviewed during the third and the start of the fourth wave of the COVID-19 pandemic, the impact of frontline/second-line conditions and gender characteristics on psychological health (in terms of depressive symptomatology and peritraumatic stress) as mediated by emotional processing.

Accordingly, it was hypothesized that being a female healthcare worker and being directly involved in treating COVID-19 patients would each be positively associated with depressive symptoms and peritraumatic distress related to COVID-19 (Hypothesis 1). Additionally, it was proposed that unhealthy emotional processing would increase the likelihood of developing depressive symptoms and peritraumatic distress due to COVID-19 among healthcare workers (Hypothesis 2). Furthermore, it was hypothesized that emotional processing would mediate the relationship between gender and frontline roles with depressive symptomatology and peritraumatic distress (Hypothesis 3).

## 2. Materials and Methods

### 2.1. Participants

A sample of 593 HCWs recruited with convenience and snowball sampling methods completed the survey between 24 March and 9 December 2021, a period which in Italy corresponds to the so-called “third wave” of COVID-19, caused by the rapid spread of the Delta variant, and the immediately following period or “fourth wave”, characterized by the spread of the so-called Omicron variant. The inclusion criteria were as follows: HCWs who, during the data collection period, were working in the COVID-19 emergency, were at least 18 years old, and had the ability to read and understand the items in the study questionnaire. This is a cross-sectional study that used a web-based survey. Healthcare students who had not yet obtained their qualification at the time of the interview were not eligible to participate in the study.

### 2.2. Procedure

This study was conducted in accordance with the Declaration of Helsinki on Ethical Principles for research involving human participants, and the study procedure received the approval of the Independent Ethical Committee of the Cagliari University Hospital [PROT. PG/2022/12537]. Participation in the study was voluntary, and no reimbursement or payment was provided to study participants. HCWs received an invitation to participate in the research via mailing lists, WhatsApp groups, and social networks. Consistent with the snowball sampling method adopted, participants were also encouraged to extend an invitation to participate in the research to other HCWs they knew. The invitation included an explanation of the main objectives of the study. Before completing the questionnaires, participants read a participant information sheet and provided digital informed consent. Then, they completed the online survey that was implemented with Google Forms.

### 2.3. Measures

#### 2.3.1. Sociodemographic and Background Variables

An ad hoc questionnaire was created to collect basic information regarding the participants, including age, gender, data such as practiced healthcare profession, position as frontline or non-frontline operator in the COVID-19 emergency, type of healthcare structure, and working region and environment of the subject.

#### 2.3.2. Emotional Processing

The model of the *emotional processing scale* (EPS) [30] focuses on assessing the style of emotional processing that the individual generally uses to metabolize the impact of life events. The EPS is a self-report measure that assesses emotional processing styles with 25 items. Subjects rate, on a 10-point response scale (from 0 = *strongly disagree* to 9 = *strongly agree*), the extent to which each of the 25 statements applies to how they felt or acted during the previous week. The questionnaire assesses five subscales, each with five items, corresponding to the following five emotional processing styles: *suppression of emotions* (excessive control of emotional experience and expression), *signs of unprocessed emotions* (intrusive and persistent emotional experiences), *unregulated emotions* or *controllability of emotions* (inability to control one’s emotions), *emotions avoidance* (avoidance of negative emotional triggers), and *impoverished emotional experience* (detached experience of emotions due to poor emotional understanding). For each of the five subscales, a higher score indicates worse emotional processing. The EPS also provides an overall score, with the highest values indicating more dysfunctional emotional processing. The EPS showed excellent internal consistency values for the full scale and fair to good values for the subscales (Cronbach’s alphas ranged from 0.67 to 0.92) [30,40,41].

#### 2.3.3. COVID-19 Peritraumatic Distress Symptoms

The *COVID-19 Peritraumatic Distress Index* (CPDI) [42,43] is a self-report questionnaire specifically designed to detect peritraumatic psychological distress during the COVID-19 pandemic. It consists of 24 items that investigate the frequency of anxiety, depression, cognitive change, avoidance, and compulsive behavior, specific phobias, physical symptoms, and loss of social functioning in the past week. The subject is asked to evaluate, on a 5-point Likert-type scale (from 0 = *not at all* to 4 = *extremely*), how often each of the experiences described has happened to him/her in the last seven days. The possible range of scores is 0 to 100, with a score below 28 indicating no distress, between 28 and 51 indicating mild to moderate distress, and ≥52 indicating severe distress. Studies examining the psychometric properties of CPDI reported good to excellent values of internal consistency (Cronbach’s alpha ranged from 0.89 to 0.95) [42,43].

#### 2.3.4. Depressive Symptoms

The *Center for Epidemiological Studies-Depression Scale* (CES-D) [44,45] is a self-report questionnaire widely used for the screening of depressive symptoms, composed of 20 items. The subject is asked to evaluate, on a 4-point Likert-type response scale, how he/she has felt in the last week. The questionnaire measures the symptoms defined by the American Psychiatric Association’s Diagnostic and Statistical Manual of Mental Disorders [46] in relation to major depressive disorder. The possible range of scores is from 0 to 60, with higher scores indicating the presence of multiple symptoms. The CES-D was found to have acceptable reliability (Cronbach’s alpha ranged from 0.85 to 0.95) [47].

### 2.4. Statistical Analyses

All analyses were made with RStudio 2022.12.0 and JASP 0.17.1 software [48]. Since the survey was created with Google Forms, we could set each item of the questionnaire as mandatory for completion of the survey, and this resulted in no missing values.

Descriptive statistics (medians, means, standard errors of the means, standard deviations, skewness, kurtosis) and reliabilities (Cronbach’s alphas and McDonald’s omegas) were estimated for each psychological scale. For interpretation of the results, we have referred to criteria well-established in the literature. Acceptable skewness and kurtosis values should be included between ±2 [49]. With regard to Cronbach’s alpha values, we considered the criteria proposed by Kline [50] (α ≥ 0.90 excellent; 0.70 ≤ α < 0.90 good; 0.60 ≤ α < 0.70 acceptable; 0.50 ≤ α < 0.60 poor; α < 0.50 unacceptable). Bivariate correlations were estimated between psychological scales. With regard to Pearson correlation values, we considered the criteria proposed by Cohen [51] (*r* < 0.30 = low; 0.30 ≤ *r* < *0*.50 = moderate; *r* ≥ 0.50 = high).

To test the connections between gender, frontline, emotional processing style, depression, and peritraumatic distress for COVID-19, we developed structural equation models to test mediation analysis. We developed a model for testing the mediation effect of emotional processing style on the relationships between gender, frontline, and depression (Figure 1) and a model for testing the mediation effect of emotional processing style on the relationships between gender, frontline, and peritraumatic distress (Figure 2). We used dummy coding for gender (males = 0; females = 1) and frontline (no = 0; yes = 1) variables. To evaluate the goodness of fit of the mediation models implemented, we considered the criteria suggested by Schermelleh-Engel et al. [52] as indices of a close fit (0 < χ^2^ < 2, 0 < RMSEA < 0.05, 0.97 < CFI < 1, 0.95 < TLI < 1, 0 < SRMR < 0.05). We also calculated the Akaike Information Criterion (AIC) and the Bayesian Information Criterion (BC) because these indexes, together with the CFI and TLI, are used to compare different structural models.

In addition, in case of significant mediation effects of emotional processing style, we analyzed with Bayesian regression models the predictive validity of EPS subscales (suppression, unprocessed, controllability, avoidance, and experience) to identify the styles that have heavier weight in affecting depression and peritraumatic distress. We estimated the Bayes factor (BF_01_) of each regression model across sets of criteria. We defined three groups of BF_01_. BF_01_ ≥ 3 indicates strong evidence in favor of the alternative hypothesis (H_1_); BF_01_ ≤ 1/3 indicates strong evidence in favor of the null hypothesis (H_0_); and 1/3 < BF_01_ < 3 indicates inconclusive evidence in favor of H_0_ or H_1_. For a correct interpretation of Bayesian regression, we have to add that BF_01_ indicates the prediction validity of the regression model when a particular predictor is omitted. If omitting a particular predictor generates a BF_01_ ≤ 1/3, this means that it is a fundamental indicator, and its omission reduces the prediction validity of the model. Therefore, low BF_01_ values indicate high importance of predictors.

Figure 1 shows the block diagram representing the statistical methods and stages applied to this study.

## 3. Results

### 3.1. Sociodemographic Characteristics and Descriptive Statistics

The sample is entirely composed of HCWs whose workplace is in Sardinia (Italy). A total of 77.6% of the participants were women; their age ranged between 19 and 70 years (*M* = 41.06, *SD* = 10.54). A total of 51.4% were nurses, 65.1% were second-line health workers in the COVID-19 emergency, and 73.0% worked in a public health facility. Table 1 shows sociodemographic and job characteristics of the study sample.

Table 2 presents the descriptive statistics of all the scales used in the data analyses.

### 3.2. Mediation Analysis

#### 3.2.1. Correlations Among the Variables

Table 3 shows the bivariate correlations between emotional processing (EPS total score), depressive symptoms (CES-D), and peritraumatic distress due to COVID-19 (CPDI). As can be seen, these correlations are all significant. Specifically, unhealthy emotional processing (EPS) showed high and positive correlations with both depressive symptomatology (CES-D) (*r* = 0.708, *p* < 0.001) and peritraumatic distress (CPDI) (*r* = 0.636, *p* < 0.001. There was also a high and positive correlation between depressive symptomatology and peritraumatic stress (*r* = 0.803, *p* < 0.001).

Table 4 shows the bivariate correlations between EPS subscales, which are all significant, positive, and ranging from moderate to high (*r* values ranging between 0.479 and 0.770, *p* < 0.001).

#### 3.2.2. Mediating Models Testing

An early model of mediation was intended to test if emotional processing mediates the relationship between gender, frontline, and depression. The results of the path model are shown in Figure 2. It shows both direct and indirect effects of gender and frontline versus non-frontline experience on depressive symptomatology through the mediation of emotional processing. The path model showed a good model fit (χ^2^ = 2.452, *df* = 1, *p* = 0.117, CFI = 0.997, TLI = 0.980, AIC = 5207.60, BIC = 5264.61, RMSEA = 0.049, 90% RMSEA = 0–0.132, SRMR = 0.018). Direct and indirect path results highly confirmed our hypotheses.

Table 5 reveals that, with regard to the total effect, gender positively predicts depressive symptomatology (*β*′ = 0.186, *z* = 4.618, *p* < 0.001). Analyzing the indirect effects, results reveal that emotional processing significantly mediates (*ab* = 0.092, *z* = 3.321, *p* = 0.001 [95% *CI*, 1.001 to 4.085]) the relationship between gender and depression. Gender positively affects emotional processing (*β*′ = 0.132, *z* = 3.261, *p* = 0.001), and emotional processing, in turn, positively affects depressive symptomatology (*β*′ = 0.696, *z* = 23.886, *p* < 0.001). In addition, the results suggest that even after accounting for the mediating role of emotional processing, gender still has a significant positive direct effect on depressive symptomatology (*β*′ = 0.094, *z* = 3.241, *p* = 0.001). It is estimated that emotional processing accounts for 49.46% of gender’s effect on depressive symptomatology.

Table 5 also shows that the total effect of the frontline experience on depressive symptoms do not appear to be significant (*β*′ = 0.075, *z* = 1.865, *p* = 0.062); however, emotional processing significantly mediates (*ab* = 0.070, *z* = 2.459, *p* = 0.014 [95% *CI*, 0.342 to 3.031]) the relationship between the frontline role in the COVID-19 emergency and depressive symptomatology. Frontline condition positively affects emotional processing (*β*′ = 0.100, *z* = 2.472, *p* = 0.013), and emotional processing, in turn, positively affects depressive symptomatology (*β*′ = 0.696, *z* = 23.886, *p* < 0.001). The direct effect of the frontline experience on depressive symptoms was not significant (*β*′ = 0.005, *z* = 0.186, *p* = 0.852). Emotional processing was estimated to account for 93.33% of the frontline experience’s influence on depressive symptomatology.

In summary, there are significant direct connections between gender, frontline role, and emotional processing, and significant direct connections between gender, frontline role, emotional processing, and depression. Female HCWs have higher levels of depression and higher difficulty in processing emotions. Frontline HCWs have higher difficulty in processing emotions.

Indirect connections between gender and depression are significant, and indirect connections between frontline and depression are significant as well. Thus, our results suggest that the female gender appears to be a significant predictor of the probability of developing depressive symptoms in our sample of HCWs. Furthermore, women tend to have more problematic emotional processing than their male counterparts, which, in turn, results in greater depressive symptomatology. Moreover, frontline HCWs tend to have more problematic emotional processing than their non-frontline counterparts, which, in turn, results in greater depressive symptomatology.

A second model of mediation was intended to test if emotional processing mediates the relationship between gender, frontline, and peritraumatic distress specific to COVID-19. The results of the path model are shown in Figure 3. It explains both direct and indirect effects of gender and frontline versus non-frontline experience on peritraumatic distress through the mediation of emotional processing. The path model showed a good model fit (χ^2^ = 2.452, *df* = 1, *p* = 0.117, CFI = 0.996, TLI = 0.973, AIC = 5316.60, BIC = 5.373.06, RMSEA = 0.049, 90% RMSEA = 0–0.132, SRMR = 0.018). The identical 90% RMSEA confidence intervals in both mediator models, despite differing by a single variable, suggest that this variable does not significantly affect the model’s overall fit. Direct and indirect path results highly confirmed our hypotheses.

Table 6 reveals that, as regards the total effect, gender positively predicts peritraumatic distress (*β*′ = 0.170, *z* = 4.221, *p* < 0.001). Analyzing the indirect effects, results reveal that emotional processing significantly mediates (*ab* = 0.082, *z* = 3.217, *p* = 0.001 [95% *CI*, 1.144 to 4.711]) the relationship between gender and peritraumatic distress. Gender positively affects emotional processing (*β*′ = 0.132, *z* = 3.261, *p* = 0.001), and emotional processing, in turn, positively affects peritraumatic distress (*β*′ = 0.624, *z* = 19.547, *p* < 0.001). In addition, the results suggest that even after accounting for the mediating role of emotional processing, gender still has a significant positive direct effect on peritraumatic distress (*β*′ = 0.088, *z* = 2.770, *p* = 0.006). Emotional processing was estimated to account for 48.23% of gender’s effect on peritraumatic distress. Our results suggest that female gender appears to be a significant predictor of the probability of developing peritraumatic distress in our sample of HCWs. Furthermore, women tend to have more problematic emotional processing than their male counterparts, which, in turn, results in greater peritraumatic distress.

Table 6 also shows that the total effect of the frontline experience on peritraumatic distress does not appear to be significant (*β*′ = 0.070, *z* = 1.738, *p* = 0.082); however, emotional processing significantly mediates (*ab* = 0.062, *z* = 2.453, *p* = 0.014 [95% *CI*, 0.390 to 3.494]) the relationship between the frontline role in the COVID-19 emergency and peritraumatic distress. Frontline condition positively affects emotional processing (*β*′ = 0.100, *z* = 2.472, *p* = 0.013), and emotional processing, in turn, positively affects peritraumatic distress (*β*′ = 0.624, *z* = 19.547, *p* < 0.001). The direct effect of frontline experience on peritraumatic distress was not significant (*β*′ = 0.008, *z* = 0.242, *p* = 0.808). It is estimated that emotional processing accounts for 88.57% of the frontline experience’s effect on peritraumatic distress. These results suggest that frontline HCWs tend to have more problematic emotional processing than their non-frontline counterparts, which, in turn, results in greater peritraumatic distress.

In summary, there are significant direct connections between gender, frontline role, and emotional processing, and significant direct connections between gender, frontline role, emotional processing, and peritraumatic distress. Female HCWs have higher levels of peritraumatic distress and higher difficulty in processing emotions. Frontline HCWs have higher difficulty in processing emotions. Indirect connections between gender and peritraumatic distress are significant, and indirect connections between frontline role and peritraumatic distress are significant as well. Thus, our results suggest that female gender appears to be a significant predictor of the probability of developing peritraumatic distress specific to COVID-19 in our sample of HCWs. Furthermore, women tend to have less healthy emotional processing than their male counterparts, which, in turn, results in greater peritraumatic distress. Moreover, frontline HCWs tend to have more problematic emotional processing than their non-frontline counterparts, which, in turn, results in greater peritraumatic distress.

#### 3.2.3. Bayesian Regression

Table 7 shows the results of the Bayesian regression models testing the predictive validity of EPS subscales (experience, avoidance, controllability, unprocessed, and suppression) conducted to identify the emotional processing styles that have a heavier weight in affecting depression and peritraumatic distress. As shown in Table 7, we estimated the Bayes factor (BF_01_) for both CES-D and CPDI regression models, which revealed that the removal of experience, controllability, and unprocessed EPS subscales reduced the prediction validity of the models (BF_01_ < 1/3). These results suggest that, in our sample of HCWs, the detached experience of emotions due to poor emotional understanding, difficulty in controlling externalizing emotions, and intrusive and persistent emotional experience are the most valid predictors of depression and COVID-19 peritraumatic distress.

## 4. Discussion

International studies have identified symptoms of psychological distress among medical and paramedical personnel during the COVID-19 pandemic. However, our literature review found a gap in studies evaluating how healthcare workers’ (HCWs) ability to process emotions influenced their psychological health. This study, conducted in Italy during the third and partly the fourth COVID-19 waves, aimed to address this gap by examining how frontline/non-frontline status and gender impacted psychological health (depressive symptoms and peritraumatic distress) mediated by emotional processing.

Our findings indicate that female HCWs and those working frontline are at higher risk for depressive symptoms and peritraumatic distress, supporting Hypothesis 1. These results align with other studies [7,11,22,23,24,53,54]. Furthermore, HCWs with unhealthy emotional processing are more prone to these psychological issues, supporting Hypothesis 2. This aligns with the emotional processing model, which links PTSD (precursor to peritraumatic distress) and depression to impaired emotion regulation [36].

Dysfunctional emotional processing, characterized by negative attention and rumination, can establish a cycle of prolonged negative mood and depressive symptoms [37]. This is similar to the processing loop in PTSD, where peritraumatic distress is a predictor [55,56]. Rachman [31] highlighted the persistence of intrusive signs of emotional activity as a sign of unsatisfactory emotional processing. Both depression and PTSD involve avoidance, rumination, and unhealthy emotional processing, hindering the healthy processing of adverse experiences [33,36,57].

Our central finding is that emotional processing mediates the relationship between gender, frontline status, and psychological health due to COVID-19, supporting Hypothesis 3. Frontline HCWs faced multiple stressors during the pandemic, including increased working hours, risk of infection, and isolation from family, exacerbating their stress. This category of workers thus faced conditions that, at least in Italy, had never been experienced before. In such a demanding and stressful context, an individual’s internal resources—such as the ability to process emotions in a functional and healthy manner—can serve as a protective factor for the mental health of HCWs. Conversely, dysfunctional emotional processing styles, such as the suppression of emotions, avoidance of emotionally charged stimuli, and uncontrolled emotional outbursts accompanied by aggression or anger, can contribute to psychological distress and undermine the mental health of HCWs. Healthy emotional processing could serve as a protective factor, while dysfunctional styles, such as suppression or uncontrolled outbursts, promote psychological distress. Important aspects for achieving good emotional regulation, such as sleep quality, the habit of physical activity, and socialization, were significantly compromised during the COVID period [58], and we can hypothesize that this did not facilitate psychological well-being among HCWs as well as in the general population.

The gender disparity in psychological health among HCWs is also mediated by emotional processing. Reflection on these results cannot overlook the fact that the COVID-19 pandemic has contributed to making the gender inequality that afflicts our societies even more evident, with repercussions that have been more negative for women than for men in many areas. The European Institute for Gender Equality https://eige.europa.eu/topics/health/covid-19-and-gender-equality/unpaid-care-and-housewor (accessed on 3 April 2025) includes, among these areas, the deterioration of global health, the increase in unpaid work related to the care and custody of children, economic difficulties, and domestic violence. In particular, the closure of childcare facilities and schools and the long periods of distance learning have required parents to increase the time and energy spent on the custody, care, and education of their children. In contexts like those of most EU countries, including Italy, where the weight of family management is still heavily unbalanced on women, it is not surprising that the latter have been more negatively affected than their partners by the new forced distribution between work and family time imposed by the pandemic. Women spent significantly more time on unpaid care work than men during and after the lockdown, reducing or changing working hours due to the need for more time for childcare. This combination of stressors likely exceeded their ability to process emotions healthily, leading to greater psychological distress. Female HCWs, already at risk due to poor work–life balance and lower career prospects [59,60], faced even greater challenges during the pandemic, making them a particularly vulnerable group [24].

### Limitations

Our study has several limitations that suggest caution in generalizing the findings. These include the use of convenience sampling and self-report measures due to COVID-19 restrictions. The sample’s gender imbalance reflects the healthcare workforce’s demographics but may also indicate cultural reluctance among men to discuss emotional health. The cross-sectional design also limits the ability to evaluate long-term influences on psychological health. A potential additional limitation could be related to the rather wide timeframe of data collection (March–December 2021, between the third and the beginning of the fourth pandemic wave), which we set in order to obtain a numerically large sample of subjects. This may have introduced a bias, as during this period, natural changes may have occurred in HCWs (e.g., psychological changes, changes related to greater knowledge of the disease, increased adaptation, etc.) that could have interfered with the outcome.

## 5. Conclusions and Implications

The mental health of HCWs is crucial for public health as it affects individual well-being and healthcare quality. The study highlights the importance of considering emotional processing styles in understanding stress impacts and planning preventive and treatment measures. HCWs with unhealthy emotional processing are more likely to develop depressive symptoms and peritraumatic distress. The study identifies being a woman, working frontline, and maladaptive emotional processing as risk factors, underscoring the need for targeted psychological support and interventions to facilitate healthy emotional processing.

Excessive stress, as experienced by HCWs during the pandemic, impacts their psychological health and performance, affecting healthcare quality globally. Future preventive actions aimed at effectively contributing to reducing HCWs’ stress should be inspired by the literature on COVID-19, for example, studies that emphasize the importance of staying active to reduce stress and release negative emotions. Promoting regular physical activity could help HCWs to improve psychological well-being and release accumulated stress from work, especially during challenging periods such as emergencies [58]. Also important in this regard are the results obtained by Puci et al. [19], who found a marked divergence between the perceived need for psychological support expressed by HCWs during the early phase of the pandemic and the relative lack of services dedicated to this purpose among healthcare providers. The detection of this discrepancy between demand and supply should be taken seriously and should provoke a serious reflection on the mental health of HCWs. These and our findings suggest the need for psychological support programs tailored to HCWs in future health emergencies, including dedicated services and careful monitoring of the psychological well-being of these professionals, also to prevent the risk of burnout. Identifying vulnerable groups—women, frontline workers, and those with maladaptive emotional processing—can help in providing targeted psychological support to improve emotional processing and mental health.

## Figures and Tables

**Figure 1 healthcare-13-00871-f001:**
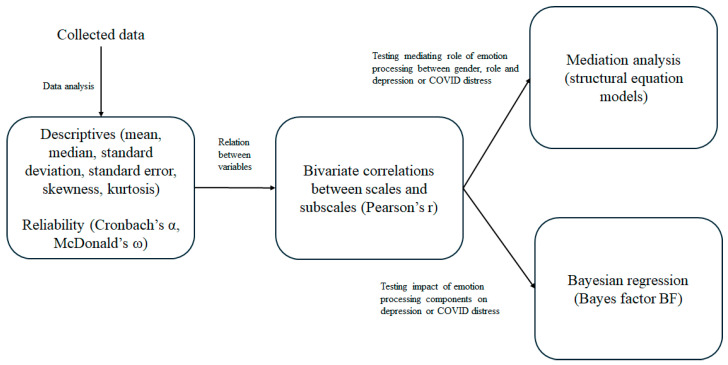
Block diagram of the statistical methods and stages applied to the study.

**Figure 2 healthcare-13-00871-f002:**
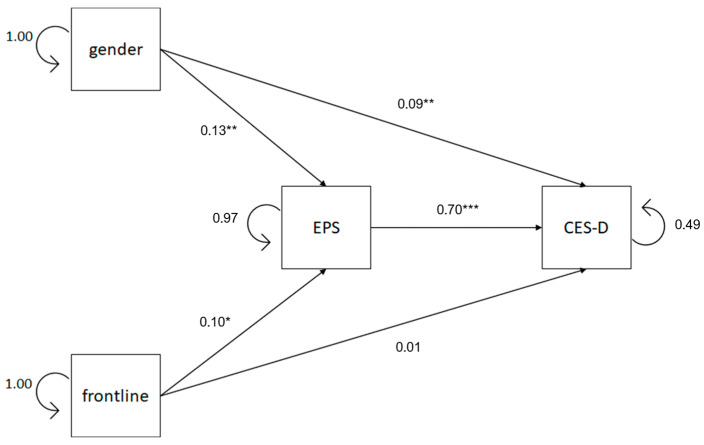
Path diagram of the mediation model between gender, frontline and EPS, and CES-D. Standardized regression coefficients and residuals are reported. Notes: EPS = Emotional processing scale, CES-D = Center for Epidemiologic Studies depression scale. * *p* < 0.05, ** *p* < 0.01, *** *p* < 0.001.

**Figure 3 healthcare-13-00871-f003:**
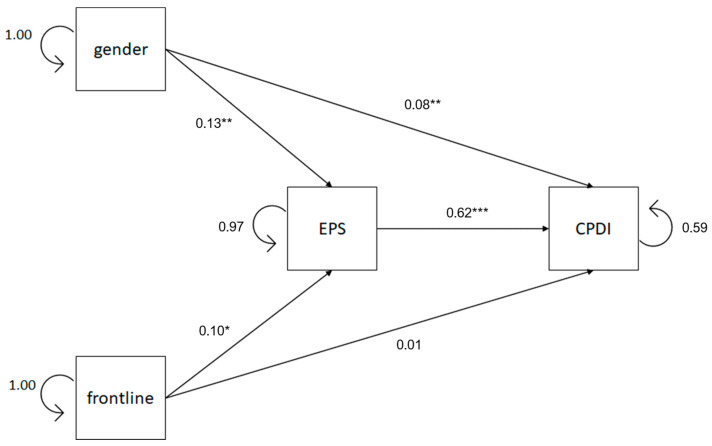
Path diagram of the mediation model between gender, frontline role, EPS, and CPDI. Standardized regression coefficients and residuals are reported. Notes: EPS = Emotional processing scale; CPDI = COVID-19 peritraumatic distress index. * *p* < 0.05, ** *p* < 0.01, *** *p* < 0.001.

**Table 1 healthcare-13-00871-t001:** Sociodemographic and job characteristics of the study sample (N = 593).

*Gender*	Frequency
Male	133 (22.4%)
Female	460 (77.6%)
*Education level*	
Secondary school (first level)	32 (5.4%)
Professional diploma	58 (9.8%)
High School Degree	141 (23.8%)
Bachelor	187 (31.5%)
Master	49 (8.3%)
Post graduate training	126 (21.2%)
*Healthcare profession*	
Nurse	305 (51.4%)
Doctor	54 (9.1%)
Healthcare Assistant	154 (26.0%)
Physiotherapist	7 (1.2%)
Speech therapist	14 (2.4%)
Psychologists	6 (1.0%)
Other	53 (8.9%)
*Workplace organization*	
Public organization	433 (73.0%)
Private organization	124 (20.9%)
Independent organization	18 (3.0%)
Other (school, home assistance, social cooperative, etc.)	18 (3.0%)
*Frontline/Second-line in COVID-19 patient’s assistance*	
Frontline	207 (34.9%)
Non-frontline	386 (65.1%)

**Table 2 healthcare-13-00871-t002:** Descriptive statistics of EPS total score, EPS subscales, CES-D, and CPDI total scores. Scale consistencies (Cronbach’s alphas and McDonald’s omegas) are reported.

		EPS Subscales						
	EPS	Suppression	Unprocessed	Controllability	Avoidance	Experience	CES-D	CPDI
Median	3.72	3.80	4.20	2.80	4.40	3.00	28.00	18.00
Mean	3.74	3.83	4.18	3.17	4.27	3.25	30.48	21.22
SE of Mean	0.08	0.09	0.11	0.09	0.09	0.09	0.47	0.61
SD	1.97	2.27	2.57	2.30	2.23	2.23	11.54	14.82
Skewness	0.12	0.10	0.09	0.61	−0.01	0.38	0.72	1.16
Kurtosis	−0.83	−0.87	−1.06	−0.46	−0.75	−0.69	−0.08	1.43
Cronbach’s alpha	0.94	0.82	0.89	0.83	0.75	0.82	0.92	0.93
McDonald’s omega	0.94	0.82	0.89	0.83	0.75	0.82	0.93	0.93

Note: SE = standard error; SD = standard deviation. Notes: EPS = Emotional processing scale; CES-D = Center for Epidemiologic Studies depression scale; CPDI = COVID-19 peritraumatic distress index; EPS subscales: suppression = suppression of emotions, unprocessed = signs of unprocessed emotions, controllability = unregulated emotions, avoidance = avoidance of emotions; experience = impoverished emotional experience.

**Table 3 healthcare-13-00871-t003:** Bivariate correlation between EPS total score, CES-D, and CPDI.

Pearson’s Correlations			
Variable	1.	2.	3.
1. Emotional processing (EPS)	—		
2. Depression (CES-D)	0.708 ***	—	
3. Peritraumatic distress (CPDI)	0.636 ***	0.803 ***	—

Notes: EPS = Emotional processing scale; CES-D = Center for Epidemiologic Studies depression scale; CPDI = COVID-19 peritraumatic distress index. *** *p* < 0.001.

**Table 4 healthcare-13-00871-t004:** Bivariate correlation between EPS subscales.

Pearson’s Correlations					
Variable	1.	2.	3.	4.	5.
1. Suppression	—				
2. Unprocessed	0.593 ***	—			
3. Controllability	0.479 ***	0.770 ***	—		
4. Avoidance	0.625 ***	0.722 ***	0.620 ***	—	
5. Experience	0.696 ***	0.712 ***	0.613 ***	0.705 ***	—

Notes: EPS subscales: Suppression = Suppression of emotions, unprocessed = signs of unprocessed emotions, controllability = unregulated emotions, avoidance = avoidance of emotions; experience = impoverished emotional experience. *** *p* < 0.001.

**Table 5 healthcare-13-00871-t005:** Results of the analyses of the first mediation model.

Effects	Path	β	SE	β′	95% Confidence Interval		z	*p*
					Lower	Upper		
Total	Gender → Depression	5.142	1.113	0.186	2.959	7.324	4.618	<0.001
	Frontline → Depression	1.817	0.974	0.075	−0.093	3.726	1.865	0.062
Indirect	Gender → Emotional processing → Depression	2.543	0.787	0.092	1.001	4.085	3.231	0.001
	Frontline → Emotional processing → Depression	1.687	0.686	0.070	0.342	3.031	2.459	0.014
Direct	Emotional processing → Depression	4.069	0.170	0.696	3.735	4.402	23.886	<0.001
	Gender → Depression	2.599	0.802	0.094	1.027	4.171	3.241	0.001
	Frontline → Depression	0.130	0.699	0.005	−1.240	1.500	0.186	0.852
	Gender → Emotional processing	0.625	0.192	0.132	0.249	1.001	3.261	0.001
	Frontline → Emotional processing	0.415	0.168	0.100	0.086	0.743	2.472	0.013

Notes: Depression as measured by CES-D; emotional processing as measured by EPS.

**Table 6 healthcare-13-00871-t006:** Results of the analyses of the second mediation model.

Effects	Path	β	SE	β′	95% Confidence Interval		z	*p*
					Lower	Upper		
Total	Gender → Peritraumatic distress	6.053	1.434	0.170	3.242	8.863	4.221	<0.001
	Frontline → Peritraumatic distress	2.181	1.255	0.070	−0.279	4.640	1.738	0.082
Indirect	Gender → Emotional processing → Peritraumatic distress	2.928	0.910	0.082	1.144	4.711	3.217	0.001
	Frontline → Emotional processing → Peritraumatic distress	1.942	0.792	0.062	0.390	3.494	2.453	0.014
Direct	Emotional processing → Peritraumatic distress	4.684	0.240	0.624	4.215	5.154	19.547	<0.001
	Gender → Peritraumatic distress	3.125	1.128	0.088	0.914	5.336	2.770	0.006
	Frontline → Peritraumatic distress	0.239	0.984	0.008	−1.689	2.166	0.242	0.808
	Gender → Emotional processing	0.625	0.192	0.132	0.249	1.001	3.261	0.001
	Frontline → Emotional processing	0.415	0.168	0.100	0.086	0.743	2.472	0.013

Notes: Peritraumatic distress as measured by CPDI; emotional processing as measured by EPS.

**Table 7 healthcare-13-00871-t007:** Bayesian factor values (BF_01_) of the regression models in relation to the omitted EPS subscales.

Omitted Predictor					
Criteria	Experience	Avoidance	Controllability	Unprocessed	Suppression
CES-D	0.040	11.207	<0.001	<0.001	12.907
CPDI	0.001	4.839	0.001	<0.001	3.771

Notes: CES-D = Center for Epidemiologic Studies depression scale; CPDI = COVID-19 peritraumatic distress index; EPS = emotional processing scale (ESP subscales: suppression = emotion suppression; unprocessed = signs of unprocessed emotion; controllability = controllability of emotion; avoidance = emotion avoidance; experience = emotional experience).

## Data Availability

The data that support the findings of this study are available from the corresponding author upon reasonable request. The data are not publicly available due to privacy restrictions.

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
