# Peer review of "Emotional Processing and Psychological Well-Being of Healthcare Workers During the COVID-19 Pandemic"

_healthcare, 2025, doi:10.3390/healthcare13080871_

Round 1
Reviewer 1 Report (Previous Reviewer 3)
Comments and Suggestions for Authors
General comment
The Authors evaluated the associations between gender, working condition and wellbeing during the COVID 19 pandemic. The study covers an interesting and actual topic, the rationale is well established and the procedures are described in details. However, there are some points that should be addressed. I hope my comments could be useful in implementing manuscript’s quality.
Introduction
Line 31-61 please synthetize the paragraph
Line 134-141 please consider to reformulate and rewrite the main aims
Methods
Line 143: please if it possible consider to add the inclusion and exclusion criteria
Discussion
Please consider to expand the whole paragraph.
Moreover, consider to add a statement regarding the practical applications derived from the current findings. For instance the contribution of physical activity with its health-related domains (quality of life, sleep) are able to ameliorate the workers’ mental-health status due to COVID-19 pandemic (e.g., Natilli M, Rossi A, Trecroci A, Cavaggioni L, Merati G, Formenti D. The long-tail effect of the COVID-19 lockdown on Italians' quality of life, sleep and physical activity. Sci Data. 2022 May 31;9(1):250)
Author Response
General comment
"The Authors evaluated the associations between gender, working condition and wellbeing during the COVID 19 pandemic. The study covers an interesting and actual topic, the rationale is well established and the procedures are described in details. However, there are some points that should be addressed. I hope my comments could be useful in implementing manuscript’s quality."
Response to the general comment: We would like to thank the Reviewer 2 for the appreciation of our manuscript and for giving us the opportunity to improve our work through their suggestions and comments.
Comment 1: “Line 31-61 please synthetize the paragraph”
Response to comment 1: As suggested by the Reviewer, we have summarized the paragraph, which now includes lines from 31 to 44:
“Italy was the first European country to experience the spread of SARS-CoV-2, beginning in January 2020 with a couple of tourists from Wuhan, China, followed by the first local cases in February [1]. As of February 2025, Italy has reported 27,192,437 cases, with 513,865 among healthcare workers (Italian National Institute of Health, https://www.iss.it/coronavirus, accessed on February 21, 2025). The spread was uneven, with northern regions more affected than the south, including Sicily and Sardinia. Sardinia, characterized by geographical isolation, limited connections, and low population density, seemed to have some protective factors. However, its aging population and strong tourism industry made it vulnerable, especially during summer, when tourist influxes exacerbated infection spread. Sardinia's pandemic timeline mirrored Italy’s, with a lockdown starting in March 2020 and a significant second wave in October 2020. The third wave (February-May 2021) saw Sardinia's situation align with the rest of Italy, aided by vaccination efforts. In the summer of 2021, tourist movement led to another peak, followed by a fifth wave (November 2021-February 2022) driven by the Omicron variant, which was less severe but highly transmissible [2].”
Comment 2: “Line 134-141 please consider to reformulate and rewrite the main aims”
Response to comment 2: We thank you for suggesting that we rephrase and rewrite the main hypotheses of the study; this section now seems improved. We hope you agree with this revision (see lines 133 – 140 in the revised version).
“Accordingly, it was hypothesized that being a female healthcare worker and being directly involved in treating COVID-19 patients would each be positively associated with depressive symptoms and peritraumatic distress related to COVID-19 (Hypothesis 1). Additionally, it was proposed that unhealthy emotional processing would increase the likelihood of developing depressive symptoms and peritraumatic distress due to COVID-19 among healthcare workers (Hypothesis 2). Furthermore, it was hypothesized that emotional processing would mediate the relationship between gender and frontline roles with depressive symptomatology and peritraumatic distress (Hypothesis 3)”.
Comment 3: “Methods. Line 143: please if it possible consider to add the inclusion and exclusion criteria”
Response to comment 3: You can find the inclusion and exclusion criteria in lines 147 to 152.
“The inclusion criteria were: HCWs who, during the data collection period, were working in the Covid-19 emergency, being at least 18 years old, and the ability to read and understand the items in the study questionnaire. This is a cross-sectional study and used a web-based survey. Healthcare students who had not yet obtained their qualification at the time of the interview were not eligible to participate in the study.”
Comment 4: “Discussion. Please consider to expand the whole paragraph”
Response to comment 4: We have decided to expand the discussion by including some reflections on the Italian context that, due to length constraints, were not included in the first version of our article. Thanks to your request, we were able to incorporate discussion points that we consider central. See lines from 458 to 466, and lines from 473 to 487.
Comment 5: “Moreover, consider to add a statement regarding the practical applications derived from the current findings. For instance the contribution of physical activity with its health-related domains (quality of life, sleep) are able to ameliorate the workers’ mental-health status due to COVID-19 pandemic (e.g., Natilli M, Rossi A, Trecroci A, Cavaggioni L, Merati G, Formenti D. The long-tail effect of the COVID-19 lockdown on Italians' quality of life, sleep and physical activity. Sci Data. 2022 May 31;9(1):250)”
Response to comment 5: Thank you for this observation, which allows us to expand on an important aspect, drawing inspiration from an interesting piece of work. We referred to these aspects, citing the work you suggested, in lines 460 - 464 “Important aspects for achieving good emotional regulation, such as sleep quality, the habit of physical activity, and socialization, were significantly compromised during the Covid period [60], and we can hypothesize that this did not facilitate psychological well-being among HCWs as well as in the general population”. And in lines 495 to 500 “Future preventive actions aimed at effectively contributing to reducing HCWs' stress should be inspired by the literature on COVID-19, for example, studies that emphasize the importance of staying active to reduce stress and release negative emotions. Promoting regular physical activity could help HCWs to improve psychological well-being and release accumulated stress from work, especially during challenging periods such as emergencies [60].”
Reviewer 2 Report (Previous Reviewer 2)
Comments and Suggestions for Authors
The authors have made a substantial revision of the article, with improvements visible and identified in the document. Regarding the observations and doubts previously identified and registered, an effort in the writing and corrections of the paper is noticeable. Given the improvements in relation to the registered questions, and taking into account this revision, I understand that the paper meets the requirements for publication without resubmitting it for review.
NOTE: I leave to the consideration of the authors the recommendation to add a block diagram defining the stages and statistical methods applied, for a better clarity and understanding of the analysis by stage.
Author Response
General comment: “The authors have made a substantial revision of the article, with improvements visible and identified in the document. Regarding the observations and doubts previously identified and registered, an effort in the writing and corrections of the paper is noticeable. Given the improvements in relation to the registered questions, and taking into account this revision, I understand that the paper meets the requirements for publication without resubmitting it for review.”
Response to the general comment: We thank you for your comments on our revised work and for making a significant contribution to improving our manuscript in terms of completeness and clarity.
Comment 1: “NOTE: I leave to the consideration of the authors the recommendation to add a block diagram defining the stages and statistical methods applied, for a better clarity and understanding of the analysis by stage.”
Response to comment 1:
After careful consideration, we followed your suggestion to include a new figure (see lines 247 - 253) with the block diagram of the statistical procedure. Consequently, we have changed the numbering of the other figures already present and their corresponding references in the text. These changes are highlighted in the text.
Round 2
Reviewer 1 Report (Previous Reviewer 3)
Comments and Suggestions for Authors
I do not have further comments or observations for the Authors.
This manuscript is a resubmission of an earlier submission. The following is a list of the peer review reports and author responses from that submission.
Round 1
Reviewer 1 Report
Comments and Suggestions for Authors
Busonera et al. present the results of an observational study aimed at examining the impact of gender and frontline/non-frontline status on the psychological health of a sample of Italian HCWs, as well as exploring the mediating effects of emotional processing in these relationships.
The study is interesting, however some aspects require attention, particularly in the methodology section. The analyses and models applied should be reported more clearly: there is inconsistency between the methods and the techniques described in the results.
However, it is not possible to fully evaluate the paper as the results are missing (tables and figures are absent), and it is unclear what has been conducted.
Please find below are some detailed suggestions:
Abstract and introducion
· Please add the study design and the study timeframe.
· Report the main results regarding the sample size and the measures used to address at least the primary aim
***
· If possible, include the Italian or Sardinian epidemiological context
· Lines 42-46, the paper could be introduced and discussed with reference to:
Puci MV, Nosari G, Loi F, Puci GV, Montomoli C, Ferraro OE. Risk Perception and Worries among Health Care Workers in the COVID-19 Pandemic: Findings from an Italian Survey. Healthcare (Basel). 2020 Dec 3;8(4):535. doi: 10.3390/healthcare8040535. PMID: 33287260; PMCID: PMC7761765.
·
Methods
2.1 participants:
In this section, some important information are missing and should be added:
· The study design, sampling method, and the period in which the study was conducted.
· Lines 113-116 (77.6% - Table 1...): these are results, not methods, they should be moved to the corresponding paragraph.
· Define the type of HCWs (inclusion criteria?).
Section 2.2: Procedures:
· The protocol number of the ethics committee is missing.
· The type of sampling could be included here in coherence with the information reported on recruitment (lines 123-125). Replace "people" with "participants" or "HCWs."
Section 2.3.1: Specify the timeframe during which the questionnaire was administered (period in which participants could respond).
Section 2.4: Statistics:
· Descriptive statistics and related indicators are missing.
· Lines 182-184: The model for which the goodness of fit was assessed is unclear.
· Specify the models used and report the level of statistical significance.
Results
3.1: The authors should present the characteristics of the sample in the descriptive analysis (Table 1, see previous comments).
3.2 Correlation analysis: If the questionnaire scores are not continuous measures, it would be more appropriate to use Spearman's correlation coefficients.
- Bayesian regressions are not reported in the methods; the techniques and models used are not clearly identifiable both in the methods and in the results, considering that tables and figures are missing.
Limitations
Information on the study design and sampling should also be included in the methods section.
When was the study conducted? Could the timeframe of the study introduce a potential bias?
- Discussion and conclusion sections: results (tables and figures are missing )...
Comments on the Quality of English Language
None
Author Response
RESPONSES TO REVIEWER 1’S COMMENTS
Comment 1: “The study is interesting, however some aspects require attention, particularly in the methodology section. The analyses and models applied should be reported more clearly: there is inconsistency between the methods and the techniques described in the results.
However, it is not possible to fully evaluate the paper as the results are missing (tables and figures are absent), and it is unclear what has been conducted.”
Response to comment 1: We would like to thank Reviewer 1 for the appreciation of our manuscript and for bringing, through their comments and suggestions, our attention to the unclear parts of our article, thus providing us with the opportunity to improve the description of our study.
However, we were surprised and at the same time saddened to read that “it is not possible to fully evaluate the paper as the results are missing (tables and figures are absent)”. In fact, we included tables and figures with the submission, in separate files from the main manuscript, referencing the specific tables and figures in the text. It was probably our mistake not to include the tables and figures in the main manuscript as well. In the resubmission, we therefore preferred to include everything in a single file."
Comment 2: “Please find below are some detailed suggestions:
Abstract and introducion
- Please add the study design and the study timeframe.”
Response to comment 2: Thank you for highlighting the lack of these two important pieces of information in the Abstract. We have integrated them, specifying that the study design is cross-sectional, used a web-based survey (line 10) and that the study period is March-December 2021, a time frame that in Italy coincided with the so-called 'third wave' and the start of the 'fourth wave' of COVID-19 pandemic (lines 14 -16).
Comment 3: “If possible, include the Italian or Sardinian epidemiological context”
Response to comment 3: We would like to thank you for suggesting that we include this aspect, which we believe makes our study more contextualized. We have added a new section (lines 31 - 63) where we briefly described the Italian epidemiological situation, with a focus on Sardinia, the region from which the sample of HCWs was recruited. This highlights the fact that the time period during which the sample responded to the questionnaire predominantly covers the third wave of COVID-19, but to a small extent also the fourth wave. At this point, for greater accuracy, we felt it was necessary to specify this aspect in all parts of the article where it was relevant; Abstract (lines 14 – 16), Participants (lines 146 – 149), Discussion (line 431). As for the title of the paper, we instead suggest that no reference be made to the phase of the pandemic during which the data were collected, and we recommend keeping it simply as "Emotional Processing and Psychological Well-Being of Healthcare Workers During the COVID-19 Pandemic."
Comment 4: “Lines 42-46, the paper could be introduced and discussed with reference to: Puci MV, Nosari G, Loi F, Puci GV, Montomoli C, Ferraro OE. Risk Perception and Worries among Health Care Workers in the COVID-19 Pandemic: Findings from an Italian Survey. Healthcare (Basel). 2020 Dec 3;8(4):535. doi: 10.3390/healthcare8040535. PMID: 33287260; PMCID: PMC7761765.”
Response to comment 4: We would like to thank you very much for suggesting that we consider the insights from this interesting work to improve our manuscript. We have read it with great interest and have cited it appropriately both in the Introduction (lines 81 and 86) and in the Discussion section (lines 487 - 495) of our re submitted manuscript.
Comment 5: “Methods
2.1 participants:
In this section, some important information are missing and should be added:
- The study design, sampling method, and the period in which the study was conducted.
- Lines 113-116 (77.6% - Table 1...): these are results, not methods, they should be moved to the corresponding paragraph.
- Define the type of HCWs (inclusion criteria?).”
Response to comment 5: Thanks to the observations of Reviewer 1 mentioned above, we realized that we omitted some important information related to the Method. We also agree that the socio-demographic characteristics of the sample should be moved to the Results section. Following the suggestions of Reviewer 1, we have reorganized and integrated the previously missing information into subsection 2.1 Participants. Specifically, we have included information related to the study design, sampling method, and inclusion criteria. The information regarding the timing and context of data collection was already present in our manuscript. The part rightly pointed out by the Reviewer as belonging to the results has been removed from subsection 2.1 Participants and correctly placed in subsection 3.1, now renamed “Sociodemographic Characteristics and Descriptive Statistics”. After these revisions, subsection 2.1 Participants is formulated as follows: “A sample of 593 HCWs recruited with convenience and snowball sampling methods completed the survey between 24 March and 9 December 2021, a period which in Italy corresponds to the so-called “third wave” of COVID-19, caused by the rapid spread of the Delta variant, and the immediately following period or "fourth wave," characterized by the spread of the so-called Omicron variant. The inclusion criteria were: HCWs who, during the data collection period, were working in the Covid-19 emergency, being at least 18 years old, and the ability to read and understand the items in the study questionnaire. This is a cross-sectional study and used a web-based survey” (lines 145 – 152).
Comment 6: “Section 2.2: Procedures:
- The protocol number of the ethics committee is missing.”
Response to comment 6: In the first submission of our article, we purposely omitted the details of the ethics committee, protocol, and study approval date for anonymity purposes. In the revised version, we have included only the protocol number, as requested (line 156).
Comment 7: “The type of sampling could be included here in coherence with the information reported on recruitment (lines 123-125)”
Response to comment 7: We followed the reviewer’s suggestion to reference the type of recruitment in lines 158-159, with a brief explanation of how it was carried out.
Comment 8: Replace "people" with "participants" or "HCWs."
Response to comment 8: Here too, we followed the reviewer’s suggestion and replaced “people” with HCWs (line 157).
Comment 9: “Section 2.3.1: Specify the timeframe during which the questionnaire was administered (period in which participants could respond).”
Response to comment 9: Prospective participants were not instructed to complete the questionnaire by a specific date. To ensure that they were individuals working as healthcare workers during the third wave of COVID, we asked for this information in the questionnaire and, additionally, relied on the completion dates automatically saved by Google Forms.
Comment 10: “Section 2.4: Statistics: · Descriptive statistics and related indicators are missing.”
Response to comment 10: Thank you for pointing out this oversight; we have completed the information by including the reported descriptive statistics (medians, means, standard errors of the means, standard deviations, skewness, kurtosis) (lines 211 – 212).
Comment 11: “Lines 182-184: The model for which the goodness of fit was assessed is unclear. Specify the models used and report the level of statistical significance”
Response to comment 11: We agree with the Reviewer regarding the lack of clarity in the description of the tested mediation models. We also believe that, partly, this lack of clarity is due to the disorganization of the paragraph 2.4 and, partly, to the reviewer’s impossibility to access the figures and tables that describe the analyses performed and the results obtained. In fact, in Figures 1 and 2, Tables 5 and 6, and in their respective captions, which we have now included in the article text, the tested mediation models are specified along with the levels of statistical significance. We have now revised the paragraph to present the tested models and applied analyses in a more organized and clearer manner, with in-text references to the corresponding figures and tables. Additionally, we have included these figures and tables within the text. If the lack of clarity referred to by the Reviewer concerns other elements that we have not identified, we kindly ask them to point these out to us.
Comment 12: “Results
3.1: The authors should present the characteristics of the sample in the descriptive analysis (Table 1, see previous comments).”
Response to comment 12: Following the suggestions of Reviewer 1, we have
placed the characteristics of the sample in subsection 3.1, renamed “Sociodemographic Caracteristics and Descriptive Statistics” (line 247 - 255).
Comment 13: “3.2 Correlation analysis: If the questionnaire scores are not continuous measures, it would be more appropriate to use Spearman's correlation coefficients.”
Response to comment 13: The scores of the questionnaires are all continuous quantitative measures, which is why we chose Pearson's r for the correlations.
Comment 14: “Bayesian regressions are not reported in the methods; the techniques and models used are not clearly identifiable both in the methods and in the results, considering that tables and figures are missing.”
Response to comment 14: The Bayesian regressions are reported in the subsection of Method 2.4. Statistical Analysis (lines 234 – 245). We hope that now, with the help of the tables included in the text, this section is clearer and more understandable (see Table 7). However, if the Reviewer still finds this part unclear or poorly described, we kindly ask them to let us know.
Comment 15: “Limitations
Information on the study design and sampling should also be included in the methods section.”
Response to comment 15: As suggested by Reviewer 1, we have included this information in the Methods section (see response to comment 5).
Comment 16: “When was the study conducted? Could the timeframe of the study introduce a potential bias?”
Response to comment 16: Our study was conducted between March and December 2021, a time span that covers the third wave and the beginning of the fourth wave of COVID-19. We chose a fairly wide timeframe in order to obtain a numerically significant sample. This could, in fact, introduce a potential bias, as in the meantime, there may have been natural changes concerning the professionals (psychological changes, related to greater knowledge of the disease, increased adaptation, etc.) that could have interfered with the outcome. We thank you for raising this issue, prompting a certainly constructive reflection, which we have included in the section of the manuscript dedicated to the study's limitations (lines 469 - 475).
Comment 17: “- Discussion and conclusion sections: results (tables and figures are missing )...”
Response to comment 17: We are sincerely sorry that the manuscript the Reviewers read was missing the attachments of Tables and Figures. We hope that the inclusion of these important elements in the revised text will facilitate the understanding of the analysis and results, as well as assist the reviewers in their revision work.
Reviewer 2 Report
Comments and Suggestions for Authors
Overall, the theoretical and practical input on the study is of great social contribution. This paper proposes an assessment of emotional processing styles and symptoms of depressive and peritraumatic distress during the third wave of the COVID-19 pandemic considering sociodemographic variables such as age, gender, front-line versus non front-line role in the pandemic emergency, among others. Based on the EPS, CPDI, CES-D models being validated for the prediction of the subscales with Bayesian regression models.
The document is well written and organized in general. However, it is suggested to make several specific changes in some sections of the document. Therefore, a revision is recommended considering the following points:
1. It is recommended to add a block diagram defining the stages and statistical methods applied, for greater clarity and understanding.
2. In Ln 80 the acronym PTSD is mentioned, however it has not been defined earlier in the document, so it needs to be defined from the beginning. Review each acronym under this approach.
3. In the Ln 149 - 151 Add the internal consistency of the cronbach's alpha of the EPS mentioned in references [29, 37, 38] as mentioned in the CPDI (Cronbach's alpha ranged from 0.89 to 0.95) and CES-D (Cronbach's alpha ranged from 0.85 to 0.95).
4. The tests of the mediator models in model 1 and model 2, the results in the statistic χ2 = 2.452, df = 1, p = 0.117 and χ2 = 2.452, df = 1, p = 0.117 respectively, also for the statistic RMSEA = 0.049 and SRMR = 0.018, why their behavior is equal, do you have any particular explanation, given that the conditions are different for the approach of each model?
5. What is the explanation for mentioning the results of 90%RMSEA 0-0.132 in each mediator model?
6. In Ln 274 the result in the second mediating model of 90%REMSEA 0 - 0.132 has the same meaning as 90%RMSEA 0-0.132 of the first mediating model? If yes, what does this mean?
7. I could not visualize in the whole document any table or figure with the results obtained. It did not even come as attachments to relate the results and discussions.
8. It is recommended to add the contribution of each of the authors within the study.
9. In reference 59 the year and doi were not added correctly. Likewise, in citations 27, 44, 48, 58.
Author Response
RESPONSES TO REVIEWER 2’S COMMENTS
Comment 0: “Overall, the theoretical and practical input on the study is of great social contribution. This paper proposes an assessment of emotional processing styles and symptoms of depressive and peritraumatic distress during the third wave of the COVID-19 pandemic considering sociodemographic variables such as age, gender, front-line versus non front-line role in the pandemic emergency, among others. Based on the EPS, CPDI, CES-D models being validated for the prediction of the subscales with Bayesian regression models.
The document is well written and organized in general. However, it is suggested to make several specific changes in some sections of the document. Therefore, a revision is recommended considering the following points:”
Response to comment 0: We would like to thank Reviewer 2 for the positive feedback on our work, and for the attention with which they read and commented on the manuscript. Their constructive suggestions allowed us to revise unclear and incomplete parts, enabling us to improve the quality of our article.
Comment 1: “1. It is recommended to add a block diagram defining the stages and statistical methods applied, for greater clarity and understanding.”
Response to comment 1: We believe that your request is related to the fact, pointed out by other reviewers and by yourself in a subsequent comment, that the manuscript you reviewed was lacking tables and figures (due to an accidental omission). We apologize that the absence of these elements made it difficult to understand the applied statistics and the results obtained. In this revision we have included tables and figures in the text that, we believe, make the methodological and statistical choices clearer, as well as the results obtained. Before including a block diagram, which would be an additional figure in the article, we would like to ask you whether, in your opinion, the new version of the article meets the clarity requirements, or if you still consider the inclusion of a block diagram to be necessary. We would be very grateful if you could provide us with this important feedback.
Comment 2: “In Ln 80 the acronym PTSD is mentioned, however it has not been defined earlier in the document, so it needs to be defined from the beginning. Review each acronym under this approach.”
Response to comment 2: We completely agree with this comment from Reviewer 2 and we thank them for pointing out this omission. We have now included the term Post Traumatic Stress Disorder before the acronym PTSD in the text (line 114). Additionally, we have carefully re-read the text to identify any similar oversights.
Comment 3: “ In the Ln 149 - 151 Add the internal consistency of the cronbach's alpha of the EPS mentioned in references [29, 37, 38] as mentioned in the CPDI (Cronbach's alpha ranged from 0.89 to 0.95) and CES-D (Cronbach's alpha ranged from 0.85 to 0.95).”
Response to comment 3: Thank you for highlighting this gap. We have included the range of the internal consistency values of the EPS, as reported in the referenced bibliography in the text (line 186). “Cronbach's alphas ranged from 0.67 to 0.92)”
Comment 4: “The tests of the mediator models in model 1 and model 2, the results in the statistic χ2 = 2.452, df = 1, p = 0.117 and χ2 = 2.452, df = 1, p = 0.117 respectively, also for the statistic RMSEA = 0.049 and SRMR = 0.018, why their behavior is equal, do you have any particular explanation, given that the conditions are different for the approach of each model? .”
Response to comment 4: If two mediator models have the same fit statistics—such as chi-square (χ²), root mean square error of approximation (RMSEA), and standardized root mean square residual (SRMR)—despite differing in one variable, different explanations can justify this result while confirming their validity. Fit indices such as χ², RMSEA, and SRMR reflect overall model fit rather than differences in specific paths or variables. If the modification introduced in the second model does not significantly alter the covariances among variables, the fit statistics might remain unchanged. This suggests that structural change does not disrupt the model's ability to represent the observed data. The CFI and TLI indexes, that are defined as comparison indexes because mostly used to compare different models, are not equivalent in the two models even if very similar. In fact, CFI = 0.997 and 0.996 for the first and second model and TLI = 0.980 and 0.973 for the first and second model. We can insert other indexes (AIC and BIC) that are used for comparing different structural models. Therefore, we inserted a new statement in the paper (pp. lines 230-232): “We also calculated the Akaike Information Criterion (AIC) and the Bayesian Information Criterion (BC) because these indexes, together with the CFI and TLI are used to compare different structural models”. These indexes have the function to show that the two mediating models even if very similar are not perfectly identical. In addition, both models are nearly saturated (i.e., they contain many parameters relative to the data). Therefore, changes in one variable might have a negligible impact on fit statistics. This often happens when models already explain most of the variance in the dataset.
Comment 5: “What is the explanation for mentioning the results of 90%RMSEA 0-0.132 in each mediator model?”
Response to comment 5: The RMSEA point estimate alone does not account for uncertainty. The 90% CI provides a range within which the true population RMSEA is likely to fall, giving a more reliable assessment of model fit. In this case, the lower bound is 0 (suggesting a perfect fit in the best-case scenario), while the upper bound is 0.132, indicating a potential poor fit in the worst case.
Comment 6: “In Ln 274 the result in the second mediating model of 90%REMSEA 0 - 0.132 has the same meaning as 90%RMSEA 0-0.132 of the first mediating model? If yes, what does this mean?”
Response to comment 6: The result of 90% RMSEA CI = 0 - 0.132 in the second mediating model has the same meaning as in the first mediating model. This means that both models exhibit the same level of uncertainty in their fit. If two models differ only in one variable but yield the same RMSEA CI, it implies that the change of the specific variable does not substantially impact on the overall fit of the model. In lines 350-352 we inserted a new statement to highlight this point: “The identical 90% RMSEA confidence intervals in both mediator models, despite differing by a single variable, suggest that this variable does not significantly affect the model's overall fit”.
Comment 7: “I could not visualize in the whole document any table or figure with the results obtained. It did not even come as attachments to relate the results and discussions”
Response to comment 7: We are sincerely sorry that the manuscript the Reviewers read was missing the attachments of Tables and Figures. We hope that the inclusion of these important elements in the revised text will facilitate the understanding of the analysis and results, as well as assist the Reviewers in their revision work.
Comment 8: “It is recommended to add the contribution of each of the authors within the study.”
Response to comment 8: We have now included this information, as requested by both Reviewer 2 and the Academic Editor (lines 511 – 515).
Author Contributions: Conceptualization, A.B.; methodology, A.B. and M.T.; software, M.T.; validation, M.G.; formal analysis, M.T.; investigation, I.P.; resources, A.B., M.T., I.P. and M.G.; data curation, A.B. and I.P. ; writing – original draft, A.B., M.T., I.P. and M.G.; writing – review & editing, I.P. and M.G.; visualization, A.B. and M.T.; supervision, A.B. and M.G.; project administration, A.B. and M.G. All authors have read and agreed to the published version of the manuscript.
Comment 9: “In reference 59 the year and doi were not added correctly. Likewise, in citations 27, 44, 48, 58.”
Response to comment 9: We thank the Reviewer for pointing out these inaccuracies in the references. We have corrected the incorrect elements; specifically, for references 58 and 59 (61 and 62 in the revised manuscript), we have corrected the year and DOI, and for references 27 and 48 (30 and 51 in the revised manuscript), we have corrected the DOI. As for reference 44 (47 in the revised manuscript), it originally does not have a DOI.
Reviewer 3 Report
Comments and Suggestions for Authors
General comment
Thanks to the Authors for this innovative work. They evaluated the psychological responses and gender differences in healthcare professionals during the 3rd wave COVID-19 pandemic. This is an interesting topic that deserves attention. The rationale was well established and the results clearly presented. I have comments that I hope could be useful in improving the manuscript's quality.
AUTHORS LIST AFFILIATIONS, AND ABSTRACT
please consider to be consistent with Journal's guide lines (i.e., affiliations' numbers character, unstructured abstract)
METHODS
Line 130: please consider to add more details regarding the customized questionnaire
DISCUSSION
Line 303: please consider to add a specific paragraph regarding the practical applications that healthcare professionals coud have adopted during the pandemic period to counteract emotions or distress like physical activity and sleep quality (e.g., Natilli M, Rossi A, Trecroci A, Cavaggioni L, Merati G, Formenti D. The long-tail effect of the COVID-19 lockdown on Italians' quality of life, sleep and physical activity. Sci Data. 2022 May 31;9(1):250).
Author Response
RESPONSES TO REVIEWER 3’S COMMENTS
Comment 1: “Thanks to the Authors for this innovative work. They evaluated the psychological responses and gender differences in healthcare professionals during the 3rd wave COVID-19 pandemic. This is an interesting topic that deserves attention. The rationale was well established and the results clearly presented. I have comments that I hope could be useful in improving the manuscript's quality.”
Response to Comment 1: Thank you for carefully considering our article and for commenting on its strengths and weaknesses. Thanks to your feedback, we have had the opportunity to integrate and improve our manuscript.
Comment 2: “AUTHORS LIST AFFILIATIONS, AND ABSTRACT
please consider to be consistent with Journal's guide lines (i.e., affiliations' numbers character, unstructured abstract)”
Response to Comment 2: As per your suggestion, in the revised version of the manuscript, we have made these elements more consistent with the journal's guidelines (lines 5 – 24).
Comment 3: “METHODS
Line 130: please consider to add more details regarding the customized questionnaire”
Response to Comment 3: Thank you for this suggestion; We have included all the information collected in the description of the customized questionnaire. If there is any specific aspect that we might have missed, we kindly ask you to suggest it to us explicitly.
Comment 4: “DISCUSSION
Line 303: please consider to add a specific paragraph regarding the practical applications that healthcare professionals coud have adopted during the pandemic period to counteract emotions or distress like physical activity and sleep quality (e.g., Natilli M, Rossi A, Trecroci A, Cavaggioni L, Merati G, Formenti D. The long-tail effect of the COVID-19 lockdown on Italians' quality of life, sleep and physical activity. Sci Data. 2022 May 31;9(1):250).”
Response to Comment 4: We thank you for suggesting this important point. In the Discussion section, we have included a reflection on the fact that important elements for emotional regulation, such as sleep quality and the habit of physical activity, were heavily compromised during the pandemic, and this, as highlighted in the work you suggested, certainly did not favor the psychological well-being of individuals, particularly those directly involved in facing the emergency (lines 486 - 494).